# Relationships between Risk Events, Personality Traits, and Risk Perception of Adolescent Athletes in Sports Training

**DOI:** 10.3390/ijerph19010445

**Published:** 2021-12-31

**Authors:** Chen Guo, Bingyang Xiao, Zhao Zhang, Jiahui Dong, Mei Yang, Gongbing Shan, Bingjun Wan

**Affiliations:** 1Children and Adolescent Physical Education Research Center, Shaanxi Normal University, Xi’an 710119, China; guochen-v@snnu.edu.cn (C.G.); zhangzhao2019224@snnu.edu.cn (Z.Z.); 15091670850@snnu.edu.cn (J.D.); ymty123@snnu.edu.cn (M.Y.); 2School of Psychology, Beijing Sports University, Beijing 100084, China; 3Department of Kinesiology, The University of Lethbridge, Lethbridge, AB T1K 3M4, Canada; g.shan@uleth.ca

**Keywords:** adolescent, athletes, personality, risk, sport training

## Abstract

Personality traits have close relationships with risky behaviors in various domains, including physical education, competition, and athletic training. It is yet little known about how trait personality dimensions associate with risk events and how vital factors, such as risk perception, could affect the happening of risk events in adolescent athletes. The primary purpose of this study is to investigate the prediction of risk events by regression analysis with dimensions of personality, risk perception and sports, relations between risk events, risk perception, and the facets of the personality dimensions via data collecting from 664 adolescent athletes aged 13–18 years (male 364, female 300). Secondary intent is to assess school-specific levels of training risks among sports schools, regular schools, and sports and education integrated schools. The results show that psychology events are the strongest predicted by personality traits, risk perception, and sports, followed by injury and nutrition. Emotionality has the most significant positive correlation with risk events, while other traits have a significant negative correlation with risk events, except agreeableness. The integration schools are more conducive to the healthy development of adolescent athletes’ personalities. Moreover, the research indicates that sports training can strengthen the development directions of different personality characteristics.

## 1. Introduction

Modern Risk Theory believes that there is not only material and tangible risk processes and the possibilities of harm but also cultural, spiritual, and intangible ones, and the latter are the soul of risk [1]. Risk implies possibilities, it points to things that may happen involving potential damage [2], and it is the harm that humans may face in the process of remolding themselves and changing the objective world, meanwhile, developing themselves in the process of overcoming risks [3,4]. Risk research studies in sports training studies mainly focus on sports injuries, such as head injuries and Anterior Cruciate Ligament (ACL) injuries. Chen et al. [5] conducted research on recovery from mild head injury in male athletes, making a considerable value for clinical judgment and athlete rehabilitation monitoring on head injuries; however, young male athletes as the only sample limits the general applicability of the results to female athletes. Vacek et al. [6] developed a multivariate risk models that could be used as screening tools to identify athletes who are at increased risk of suffering a non-contact ACL injury, and this has provided potential and effective prevention and screening tools for identifying athletes at increased risk of ACL injury. Previous studies not only focused on sports injuries but also paid attention to the possible risks of nutrition, psychology, and environment that adolescent athletes may face. For example, the study of Charlotte [7] showed that teammates may have a positive or negative influence on eating and exercise attitudes/behaviors of athletes. This contributes to the development of team-based interventions to reduce or prevent adolescent athletes’ disordered eating and negative exercise attitudes/behaviors; however, the study lacked more longitudinal exploration, considering the role of moderating factors, such as sport type. The factors determined by personality, such as cognition, emotion, and volition [3], have an important effect on the risks of sports training for young athletes who have not formed a complete system of self-defense experience [8,9]. There are research studies that linked personality traits to risky behaviors in a host of domains, including education [3], sports [2,10], and academia [11,12]. However, it is surprising that domestic research studies on athletes’ sports training risks are still fragmented, especially the personality and competitive risks of young athletes.

The psychological factors that constitute the athletes’ competitive ability are mainly reflected in the emotions and wills of participating in the training. The will qualities are determined by personality [13]. The trait is a basic element of personality which shows the stable and lasting behavior tendency of individuals [14]. As an important reference indicator of talent selection and development of athletes, personality has relative stability and durability, and its development is an organic unity process of stages and continuity. The development of adolescent personality is based on the premise of genetic factors, and it is constantly promoted in the process of solving contradictions, which is produced between the new demands to adapt to the objective requirements of reality and the original psychological state. In Britton’s study, the results indicate that adolescent athletes with stable emotions, extraversion, openness, and higher life satisfaction had lower stress reactivity [15]. In addition, studies also explored the predictive relationship between risk events and personality traits of adolescent athletes. Fie [16] identified a novel association between Harm Avoidance (HA) and concussion in rugby players and suggested that lower HA may lead to increased dangerous in rugby competition, influencing their concussion susceptibility. Markati et al. [17] investigated the predictive relationship among psychological (e.g., motives), situational determinants (e.g., hours of training per week and perceived volume of training), and athlete burnout symptoms (e.g., reduced sense of accomplishment) by using Canonical Correlation (CC) analysis. The results highlight burnout occurrence in adolescent athletes is a psychological risk event caused by multiple factors. Cristina et al. [18] investigated the influence relation between individual personality characteristics (e.g., self-esteem) and eating disorder risk through conducing the comparison between female gymnasts and football athletes and non-athletes. The results suggest that non-athlete female adolescents may have an enhanced risk of developing clinical eating disorders. However, larger samples and other athletes, such as males or participants from other sports, should also be considered.

Athletes’ personality is corelated to risk events and perception. To effectively control and prevent the occurrence of athletes’ risk events, it is also necessary to investigate the athletes’ perception of risks in sports training [19]. Philip Von Rosen suggests that educating athletes about how to interpret pain signals is a strategy to improve their risk perception [20]. Mayer et al. [21] found out that type of sport, perceptions of social pressure, coach’s leadership style, and athletes’ age are the social and individual determinants of the athletes’ Willingness to Compete Hurt (WCH); this finding is helpful for developing strategies of effective promotion in young athletes’ risk perception. The current consensus on the connotation of risk perception is that the estimation of the probability of risky events, risk control, and confidence under certain circumstances is an individual’s subjective judgment [22]. Based on the Information-Processing Theory, this study believes that risk perception is a process in which athletes recognize, predict, and control risk events during training and competition under the influence of personality traits, and it is also a psychological process in which risk information is processed and judged based on experiences [14,23].

According to the classification of the essential elements of sports training, the risk events of sports training are divided into five categories: injury, disease, psychology, nutrition, and environment. Correspondingly, the basic elements and premise conditions of adolescent athletes’ training of these 5 categories mainly reflected in the following 5 aspects: physical health, mental health, customs and culture, material resources, and the teaching concepts and abilities of coaches; all of them are the basic conditions of athletic training, the supporting elements of young athletes’ healthy development, and the inducement and expression of risk events (Figure 1). These 5 aspects take sports training activities as the core and constitute an entire system, as well as condition and influence each other.

The first objective of the present study is, therefore, to assess the prediction of risk events by personality traits and risk perceptions in different sports training conditions (sports events and schools). In other words, we believe that there is the possibility of mutual adaptation and selection between athletes’ personality traits and risk events, and the educational (schools) and training (sports events) environments are crucial factors in this interaction. The second objectives of this study are: (i) to investigate the specific levels of the five-factor personality traits in schools, sports, gender, age, and their relationship; and (ii) to investigate the characteristics and level of adolescent athletes’ risk perception and risk events during sports training.

## 2. Materials and Methods

### 2.1. Participants and Procedures

For acquiring the most diversity in our primary study variables, and bringing more adolescent athletes into this study, we limited our exclusion criteria. Participants inclusion criteria were as follows: (1) Athletes who enrolled in sports schools, sports and education integrated schools, or regular schools in Shaanxi, China. (2) Athletes aged from 13 to 18, as to investigate adolescent athletes in the schools mentioned above. (3) Athletes who have engaged in training for more than one year and performed more than 16 h of training per week. Eight hundred questionnaires were distributed, and 716 were retrieved, including 664 valid questionnaires. Eventually, 664 students (300 girls; 364 boys) from 13 key training institutions of youth competitive sports in Shaanxi province of China were recruited in this study. The study protocol was reviewed and approved by the Academic Committee of Shaanxi Normal University in 2021 (NO: 202116008). Before participating in this study, participants were informed that they are voluntarily recruited, their data were collected anonymously, and they could withdraw from the study anytime. Signed informed consent were provided by the participants or their parents. The investigators administered the surveys to consenting students to finish the questionnaires (duration 15–20 min) during one of their regularly scheduled classes. Those who did not attend classes completed online during the same time limitation.

### 2.2. Measures

The survey included demographic questions, the Five-Factor Personality Inventory (FFPI), the Athletic Training Risk Perception Scale (ATRPS), and the Athletes Training Risk Events Questionnaire -Adolescent (ATREQ-A).

#### 2.2.1. FFPI

The FFPI [24] is a short valid inventory that was extracted from a version of NEO PI-R. FFPI is a 50-item self-report measure designed to assess personality traits of Chinese children and adolescents, including 5 dimensions: openness (O), conscientiousness (C), extraversion (E), agreeableness (A), and emotionality (N). The values are rated by 5-point Likert-type scales (1 = strongly disagree, 5 = strongly agree), ranged from 50 to 250, with each subscale scores ranged from 10 to 50. Higher scores represent a higher level of personality trait, while the opposite is true for emotionality, with higher scores reflecting lower emotionality levels. Zou and Zhang [25] report internal consistency of the subscales of the FFPI is good, with Cronbach alpha coefficients between 0.80 and 0.85. For the present study, total internal consistency is calculated as 0.83, with good reliabilities with each subscale (α = 0.84, 0.88, 0.74, 0.85, and 0.88, respectively).

#### 2.2.2. ATRPS

The ATRPS [26] is a 15-item questionnaire designed to measure 5 dimensions of athletic training risk perception, including overall risk, severity, possibility, uncontrollability, and anxiety [19,26]. Participants were instructed to rate the degree of their agreement for each statement on a 5-point Likert scale from 1 (strongly disagree) to 5 (strongly agree). Scores on the ATRPS range from 5 to 75, with higher scores reflecting a greater degree of risk perception. The internal consistency for the present study is 0.82, and overall risk (α = 0.85), severity (α = 0.87), possibility (α = 0.87), uncontrollability (α = 0.85), and anxiety (α = 0.80) show good reliabilities.

#### 2.2.3. ATREQ-A

The ATREQ-A [19] is a 65-item questionnaire designed to assess risk events happened among athletes training of adolescent in China, mainly based on the personality characteristics of adolescent athletes and the incidence of sports training risk events [6,16,27,28]. The ATREQ-A consists of 5 dimensions, injury, disease, psychology, nutrition, and environment [19]. Injury refers to which most likely and frequently happened in underage athletes [19]. Disease covers the common developmental diseases, as well as acute and chronic diseases, caused by sports injuries in the adolescent athlete [19]. Psychology is defined as unhealthy, emotional, and psychological risks arisen from competitive sports training [19]. Nutrition refers to harmful consequences of malnutrition or overnutrition, as well as unhealthy lifestyle habits (such as alcoholism, tobacco addiction, etc.), on young athletes [19]. In addition, environment includes natural environmental disasters, training environmental risks, and social environmental risks [19]. Participants were instructed to rate the frequency to each risk event that happened among their training on a 5-point Likert scale from 1 (never) to 5 (always). Values for the ATREQ-A range between 65 and 325, with higher values reflecting more risk events occurrence. For the current study, the ATREQ-A reports excellent internal consistency, with a Cronbach alpha of 0.91, and good internal consistency for subscale, injury (α = 0.92), disease (α = 0.78), psychology (α = 0.80), nutrition (α = 0.80), and environment (α = 0.83). The results of the confirmatory factor analysis showed that the validity of the structure of the questionnaire met the measurement criteria (χ^2^/df = 1.25, RMSEA = 0.06, GFI = 0.84, NFI = 0.85, CFI = 0.96, IFI = 0.97).

### 2.3. Statistical Analysis

The results were analyzed by using the Statistical Package for the Social Sciences (SPSS version 24.0). Variables were screened for normality and alpha reliability coefficients (α), ANOVA and *T*-test were used to exam differences in personality traits, risk perception, and risk events by gender, age, schools, and sports, and then Pearson bivariate correlation coefficients were computed. The relative prediction of each predictor dimension (including personality traits, risk perception, and sports) on risk events was assessed using linear regression, with each dimension entered simultaneously. The level of significance was set at *p* < 0.05.

## 3. Results

### 3.1. Demographic Characteristics of the Participants

Table 1 shows the descriptive statistics of the participants (*n* = 664). The mean age of the participants was 15.46 years (±SD 1.58). In these institutions, 6 sports schools (393 students, 59.19%), 3 sports and education integrated schools (38 students, 5.72%), and 4 regular schools (233 students, 35.09%) were involved. The sports that the participants engage in include track and field (156 students, 23.49%), soccer (132 students, 19.88%), basketball (126 students, 18.98%), volleyball (115 students, 17.32%), and the sports graded by body weight (135 students, 20.33%), including wrestling, Sanshou, Judo, and boxing.

### 3.2. Common Method Biases Test

Since the data in this study were collected by questionnaires and scales, to eliminate common method biases, Harman single factor test was used to inspect common method biases [29] based on the tools proposed by predecessors to control common method deviation [30,31,32]. The results show that there are 15 factors with eigenvalues greater than 1, and the variation explained by the first factor is 26.51%, which is less than 40% of the critical standard criterion, proving that the deviation of the common method is not obvious [32].

### 3.3. Preliminary Analyses

Descriptive statistics for each of the measured variables was provided in Table 2 and Table 3. ANOVA and *T*-test were adopted to determine differences in personality traits, risk perception, and risk events as a function of age, gender, schools, and sports (Table 3). Results reveal that there are significant differences in the personality traits in terms of sports, gender, and schools, whereas only conscientiousness shows significant differences in age. Follow-up scores of risk perception show significant differences in sports and schools, but few in age and gender. For risk events, the scores of diseases and psychology are relatively high, while the nutrition events are relatively low. Significant differences are confirmed for injury, disease, psychology, and nutrition in different sports, gender, and schools, while, in different age groups, only environmental events show significant differences.

Regarding each variable, correlation coefficients (r) are presented in Table 4, in which report statistically significant correlations among personality traits, risk perception, and risk events. Most notably, emotionality has a positive correlation with risk perception and risk events, which indicates that participants have higher values on emotionality tend to report higher risk perception and more risk events. Extraversion presents negative relationships with risk perception which means participants who have higher scores on extraversion may be more likely to have lower risk perception levels. Moreover, risk perception shows positive correlations with injury and psychology events, which indicates that participants with higher levels of risk perception may be more vulnerable to injury events and psychology events. The differences in emotionality traits of adolescent athletes in sports, gender, and schools, and their correlation with risk events and risk perception are the most significant and the most obvious characteristics.

Based on the results of the correlation between personality traits and specific events of five kinds of risk events, Figure 2 shows 24 detailed risk events which approach statistical significance with personality traits. This indicates that a personality trait is correlated with multiple specific risk events, proves that no matter which obvious personality traits adolescent athletes have, their personality develops as a whole, especially for the athletes whose personality structure is not yet stable enough, and sports training activities and environmental factors can easily affect their personality traits, which also explains the risk events that are the results of multiple personality incentives.

### 3.4. Regression Analysis

Finally, to investigate the personality traits, risk perception, and sports that predict risk events, regression analyses are performed simultaneously, entering as predictors only for those variables that were correlated statistically and predicted (*p* < 0.05) risk events.

According to Table 5 and Table 6, the regression model of each risk event is statistically significant, and the Durbin Watson values (D-W value) of each model are around 2, indicating that the autocorrelation of each model is weak. There is no obvious correlation between residuals, and the regression equations are dependable. Moreover, the regression coefficient of each predictor is significant. In the multiple collinearity test, the tolerance is about 0.5–0.9, and the expansion factor values are all less than 3, revealing that the collinearity among the variables is relatively weak, and the coefficients in the regression equations are relatively stable. In the multiple regression model, the absolute value of the standardized coefficient Beta represents the contribution of the independent variable to the dependent variable after eliminating the influence of the measurement unit. The standardized coefficient Beta of the independent variable predicted by this study represents the degree of its influence on the dependent variable risk events, and the absolute value of the Beta reflects the contribution of this independent variable to the dependent variable [33].

Psychology is the strongest factor of risk events predicted by personality traits and risk perception, followed by disease and injury. However, nutrition is not significantly predicted by personality but by factors of risk perception. We find that emotionality is the strongest trait in predicting psychology positively, following agreeableness, while extraversion has the most negative association with psychology. Concretely, the contribution rate of personality traits to predicting psychological risk events is 57%, of which 17% are emotionality (positive), 25% are extraversion (negative), and 15% are agreeableness (positive). Our results also reveal that emotionality, uncontrollability, and possibility are positively associated with injuries among volleyball and body weight graded sports. More specifically, for adolescent athletes playing volleyball and body weight graded sports, emotionality trait, uncontrollability, and possibility could predict 22% of injury. Moreover, facet predictors of personality in disease are emotionality and openness, whereas factors of risk perception are not significant. Meanwhile, when entered simultaneously, predictors of risk perception in nutrition are overall risk and severity; however, personality traits are not significant to predict nutrition events. To conclude, we admit that traits of five-factor personality have a strong relation with risk events of adolescent athletes; emotionality and agreeableness could positively predict risk events, whereas openness, extraversion could predict negatively. Further, risk perception could also predict risk events positively.

## 4. Discussion

This study contributes to current knowledge of adolescent athletes training by revealing the personality and risk perception factors that are related to risk events, as well as characteristics of personality traits and risk perception. Higher emotionality and risk perception are predictors of risk events, especially for psychology events. Higher openness or extraversion predicts lower disease or/and psychology. Another key discovery is that risk events in sports and education integrated schools are lower than sports schools and regular schools, which indicates integrated schools may be the most proper environments for the healthy development of teenage athletes. The findings of the current study indicate that the adolescent athletes show higher agreeableness and lower emotionality, which are related to the socialization of adolescent athletes [34,35] and the construction of the athletic training environment [3]. Agreeableness is a pro-social trait expressed in positive words, such as helpfulness and consideration, recognized as interpersonal communication. In this study, the adolescent athletes perform well in social personality characteristics. Zou et al. [36] explains that the development of personality basically follows the principle of maturity, and such social characteristics as agreeableness will gradually increase, while neuroticism related to mental health will gradually decline; moreover, poor physical conditions could certainly affect the mental state of athletes, which, in turn, increases the risks of athletic training [6,27,28,37]. Our results show the similar characteristics, i.e., adolescent athletes have more noteworthy features in emotionality [38]. In addition, the adolescent athletes of team sports are more extroverted. Team sports require athletes to interact frequently with teammates, coaches, and even opponents. In the long-term training and competition, the team competencies and communication experiences of adolescent athletes have been improved, stimulating the desire of athletes’ self-expressiveness, as, thus, team sports athletes are more anxious and outgoing [39]. For educational environmental factors, the participants in sports schools show high emotionality, low agreeableness, and low conscientiousness, while those in regular schools show high agreeableness, high conscientiousness, and low emotionality. The competition among players in sports schools is stronger, which results in relatively weaker pro-social behaviors. Moreover, the reform of the system would lead to greater developmental risks for the athletes who study in the mode of sports schools. As the age of athletes increases, the pressures of training and competition would be induced by other negative behaviors in the environment lacking long-term guarantees, which would lead to strong anxiety, training wearied, and more. Therefore, accelerating the transformation of the sports schools’ cultivation model and updating the concepts of youth training with a concentration on overall development are particularly essential for the development of adolescent athletes and the cultivation of reserve talents.

Many studies have found significant relationships between cognitive abilities and personalities [23,40]. In the present study, it is found that the risk perception of participation is low and significantly correlated to the emotionality trait in a positive way. Studies have confirmed that, in the HEXACO personality framework, emotionality is associate with higher perceived risks [41], and the risk propensity is the result of the joint effect of both the subjects and the situational factors [41]. Based on the results, adolescent athletes generally have obvious emotionality traits, and regular schools are weaker and marginalized in the education of students’ sports risks, as, thus, the athlete’s abilities to control sports risks are even weaker. It should be noted that adolescent athletes have a strong sense of presence and care about coaches’ positive evaluation of themselves [42] and peer recognition. Therefore, their attention is mainly allocated to the improvement of performance and the recognition of others, while less attention is paid to their risk perception. Notwithstanding, the risk perception of students in integration schools is higher, and this environment may be more beneficial for athletes’ growth, with the advantages of combining the environments, resources, cultural and educational atmosphere, and role identification elements of the sports schools and the regular schools.

Athletes engaged in basketball and body weight graded sports respond to the highest level of risk perception, which is closely related to the technical and tactical characteristics of such sports, as well as the continuous strengthening of the competitive environment. Basketball sports require more intensive distance, more frequent fighting, and faster conversion between attack and defense; thus, higher of athletes’ risk cognition and self-protection awareness are required. Bodyweight graded sports, such as boxing, have direct physical contact, and athletes need to show their strength in the process of bearing micro-risk events (minor injuries, pain). Adolescent boxers have strong risk resistance, and, in the systematic training and competition, the risk perception is directly converted into their risk perception abilities [5].

Risk events are not only a reflection of athletic training risks but also a specific indicator for evaluating the systematic risks of the adolescent athletes’ development programs. The results reflect that the frequency and severity of psychology events are higher than others. Puberty is the most rapid and drastic stage of an individual’s physical and mental development, and individuals in this period need to face more complicated and intensive development tasks than other periods [36]. Non-intellectual factors, such as individual interests and emotionality, will gradually be mature; meanwhile, the interaction with their parents and peers, as well as other social relationships, will also experience a transformation, and the contradictions and conflicts in this process will increase the possibilities of psychological events, to a certain extent [43]. Judging from the current training environments and the anticipated social status, adolescent athletes still need to consider a series of problems caused by the sustainable development of athletic careers. These problems bring greater pressures on the development of young athletes, leading to the increased probability and frequency of psychological events, and the significant positive correlation between emotional personality traits and risk events would also verify this conclusion.

Our study provided the predicting outcomes of risk events by personality traits and risk perceptions in different sports training conditions (sports events and schools). However, we acknowledge that our study has several limitations. First of all, the training period of adolescent athletes selected in this study is only set as at least one year, neglecting the possible influence of different training period on adolescent athletes. For example, whether adolescent athletes with longer training years will influence risk perception due to the strengthening of risk perception brought by experience accumulation. Second, the participants are all from Shaanxi Province, China, so the data results, especially in personality traits, may have certain regional characteristics. Therefore, the results of this study have some limitations in reflecting the general personality traits of Chinese adolescent athletes.

Due to the limitations mentioned above, more improvements could be made in future studies. First, different training years group (e.g., 1 year, 2–3 years, 4–5 years or more) could be added for more detailed data comparison in order to draw stronger conclusions, as well as to reduce the influence of non-control variables on data results. Second, future research studies could consider recruiting more adolescent athletes from provinces and cities with different economic characteristics, so as to reduce the individuation of results and increase the universality.

## 5. Conclusions

The current study unveils that being risky is an obvious characteristic in the personality traits of adolescent athletes, which are mainly manifested in the diversity of personality traits in risk events. At the same time, it exerts mutual influences on the characteristics of sports, training environments, and schools [36,44,45]. As the theorists of risk believe, risk events are the events that occur under limited rationality and are easily overlooked [36,44,45]. Overall, emotionality has the strongest correlation with adolescent athletes’ training risk events and has the most significant impact on adolescent athletes’ sports risks. Athletes under the sports schools have the highest emotional quality and worst emotional stability. Moreover, the risk of personality traits is comparative since personality works in the training of adolescent athletes, and different personality traits have different degrees of performance on the same risk event. Last, but not least, emotional traits are the primary traits that affect the risks of adolescent athletes’ development, and they are more susceptible to the influences of the training environments, which may, in turn, aggravate the risks of adolescent athletes’ sports training.

## Figures and Tables

**Figure 1 ijerph-19-00445-f001:**
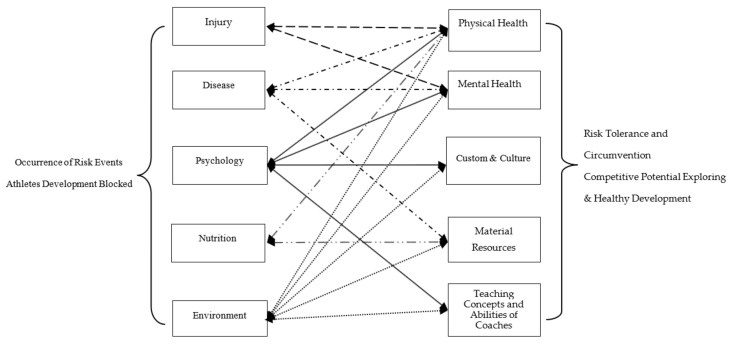
Relationships between risk events and sports training prerequisites of adolescent athletes.

**Figure 2 ijerph-19-00445-f002:**
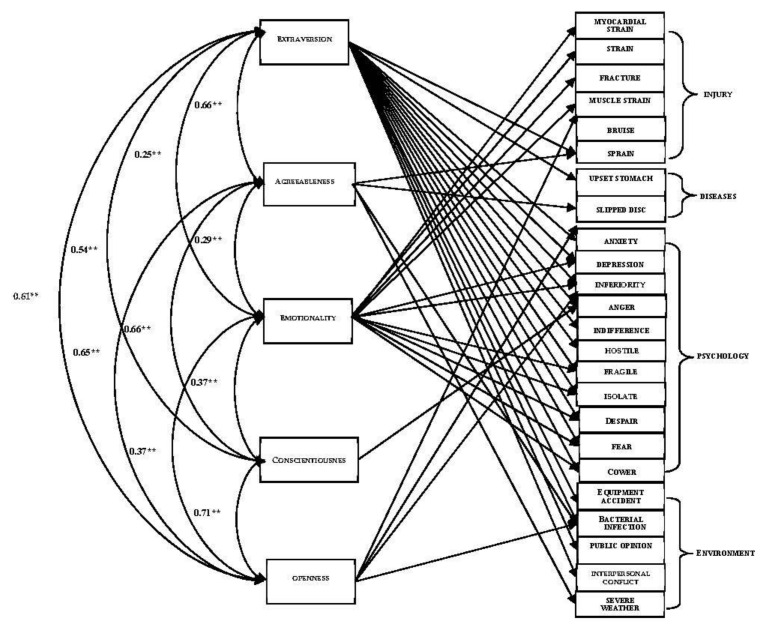
The correlation between personality traits and the structure of detailed risk events (** *p* < 0.01).

**Table 1 ijerph-19-00445-t001:** Characteristics of the study samples.

	*n*	% or Mean (SD)
Socio-demographics		
Male	364	54.82
Female	300	45.18
Mean age	664	15.46 (1.58)
13–15 years	306	46.08
16–18 years	358	53.92
Training institutions		
Sports schools	393	59.19
Integration Schools	38	5.72
Regular schools	233	35.09
Sports		
Track and field	156	23.49
Soccer	132	19.88
Basketball	126	18.98
Volleyball	115	17.32
Body weight graded sports	135	20.33

**Table 2 ijerph-19-00445-t002:** Descriptive statistics for all included variables (*n* = 664).

		M	SD	Kurtosis	Skewness
Personality Traits	Extraversion	3.72	0.72	−0.64	−0.30
	Agreeableness	3.91	0.63	−0.44	−0.42
	Emotionality	3.21	0.77	−0.23	−0.35
	Conscientiousness	3.47	0.71	−0.83	0.14
	Openness	3.58	0.69	−0.17	−0.04
Risk Perception	Overall Risk	2.46	0.90	−0.39	0.02
	Severity	2.44	0.95	−0.50	0.20
	Possibility	2.42	0.85	−0.57	0.03
	Uncontrollability	2.27	0.96	0.51	0.75
	Anxiety	2.31	1.04	−0.52	0.48
	Total	2.38	0.72	0.38	0.25
Risk Events	Injury	1.80	0.52	0.29	0.54
	Disease	1.98	0.53	0.54	0.51
	Psychology	2.10	0.53	0.55	0.56
	Nutrition	1.22	0.42	0.49	0.53
	Environment	1.84	0.44	−0.21	0.07

NOTE: M, Mean; SD, Standard Deviation.

**Table 3 ijerph-19-00445-t003:** The differences of all included variables among sports, age, gender, and schools (*n* = 664).

	Sports											Age					Gender					Schools						
	Track and Field		Soccer		Basketball		Volleyball		Body Weight Graded Sports			13–15		16–18			Male		Female			Sports Schools		Integration Schools		Regular Schools		
	M	SD	M	SD	M	SD	M	SD	M	SD		M	SD	M	SD		M	SD	M	SD		M	SD	M	SD	M	SD	
E	3.58	0.74	3.95	0.61	3.75	0.60	3.95	0.76	3.49	0.76	*F* = 12.64 *p* = 0.00 **	3.72	0.73	3.72	0.71	-	3.79	0.74	3.64	0.69	*t* = 2.67 *p* = 0.01 **	3.58	0.69	3.60	0.88	3.98	0.64	*F* = 23.85 *p* = 0.00 **
A	3.58	0.74	4.03	0.59	3.86	0.56	4.24	0.61	3.69	0.71	*F* = 16.19 *p* = 0.00 **	3.89	0.65	3.91	0.61	-	3.95	0.61	3.86	0.66	-	3.78	0.63	3.90	0.39	4.12	0.61	*F* = 22.25 *p* = 0.00 **
N	3.30	0.72	2.92	0.82	3.59	0.60	3.09	0.97	3.16	0.56	*F* = 14.69 *p* = 0.00 **	3.21	0.76	3.22	0.79	-	3.14	0.80	3.30	0.73	*t* = -2.57 *p* = 0.01 *	3.31	0.69	3.16	0.62	3.05	0.90	*F* = 8.83 *p* = 0.00 **
C	3.53	0.64	3.52	0.55	3.31	0.83	3.79	0.69	3.24	0.71	*F* = 12.30 *p* = 0.00 **	3.37	0.69	3.56	0.71	*t* = −3.54 *p* = 0.00 **	3.60	0.70	3.32	0.69	*t* = 5.06 *p* = 0.00 **	3.29	0.68	3.83	0.76	3.72	0.64	*F* = 35.77 *p* = 0.00 **
O	3.54	0.81	3.89	0.52	3.66	0.56	3.59	0.74	3.25	0.62	*F* = 16.16 *p* = 0.00 **	3.55	0.61	3.61	0.76	-	3.63	0.67	3.51	0.72	*t* = 2.37 *p* = 0.02 *	3.43	0.64	3.80	0.80	3.80	0.70	*F* = 23.61 *p* = 0.00 **
OR	2.32	0.79	2.55	0.88	2.54	1.02	2.11	0.84	2.77	0.85	*F* = 10.33 *p* = 0.00 **	2.48	0.91	2.45	0.89	-	2.48	0.89	2.45	0.92	-	2.56	0.94	2.21	0.62	2.34	0.85	*F* = 5.89 *p* = 0.00 **
S	2.33	0.77	2.61	0.95	2.94	0.98	2.16	1.10	2.59	0.88	*F* = 5.21 *p* = 0.00 **	2.45	0.98	2.43	0.91	-	2.43	0.92	2.45	0.98	-	2.50	0.94	2.45	0.69	2.34	0.98	-
P	2.38	0.72	2.46	0.91	2.72	0.88	2.10	0.66	2.41	0.96	*F* = 8.62 *p* = 0.00 **	2.29	0.84	2.53	0.85	*t* = −3.69 *p* = 0.00 **	2.42	0.84	2.41	0.87	-	2.54	0.89	2.29	0.57	2.24	0.80	*F* = 9.95 *p* = 0.00 **
U	2.06	0.73	2.14	0.84	2.62	1.10	1.96	0.61	2.58	1.21	*F* = 14.13 *p* = 0.00 **	2.20	0.89	2.33	1.02	-	2.35	0.98	2.17	0.93	*t* = 2.35 *p* = 0.02 *	2.41	1.08	2.11	0.56	2.06	0.74	*F* = 10.12 *p* = 0.00 **
ANX	2.40	0.89	2.20	1.11	2.06	1.17	2.50	0.98	2.35	1.03	*F* = 3.50 *p* = 0.00 **	2.16	1.06	2.43	1.02	*t* = −3.35 *p* = 0.00 **	2.37	1.03	2.23	1.06	-	2.30	1.09	2.58	0.60	2.27	1.00	-
I	1.71	0.50	1.76	0.42	1.78	0.56	1.85	0.53	1.93	0.56	*F* = 3.19 *p* = 0.00 **	1.77	0.45	1.83	0.57	-	1.76	0.55	1.85	0.47	*t* = −2.06 *p* = 0.04 *	1.88	0.49	1.30	0.41	1.76	0.52	*F* = 23.90 *p* = 0.00 **
D	1.93	0.66	1.88	0.38	2.00	0.54	2.04	0.48	2.04	0.51	-	1.98	0.51	1.97	0.55	-	1.18	0.50	2.18	0.50	*t* = −9.53 *p* = 0.00 **	2.06	0.54	1.33	0.36	1.95	0.45	*F* = 36.18 *p* = 0.00 **
PSY	2.04	0.63	1.95	0.49	2.09	0.37	2.21	0.48	2.22	0.58	*F* = 6.26 *p* = 0.00 **	2.13	0.49	2.08	0.57	-	2.02	0.53	2.20	0.52	*t* = −4.49 *p* = 0.00 **	2.19	0.53	1.57	0.47	2.03	0.50	*F* = 28.80 *p* = 0.00 **
NTT	1.28	0.47	1.05	0.38	1.31	0.47	1.22	0.24	1.21	0.43	*F* = 7.88 *p* = 0.00 **	1.24	0.47	1.20	0.37	-	1.20	0.41	1.24	0.43	-	1.30	0.45	0.93	0.35	1.13	0.33	*F* = 22.05 *p* = 0.00 **
EVM	1.71	0.52	1.77	0.38	1.89	0.35	1.90	0.48	1.95	0.37	*F* = 7.42 *p* = 0.00 **	1.77	0.44	1.89	0.43	*t* = −3.51 *p* = 0.00 **	1.79	0.45	1.89	0.41	*t* = −2.92 *p* = 0.00 **	1.89	0.41	1.25	0.30	1.85	0.42	*F* = 42.44 *p* = 0.00 **

NOTE: M, Mean; SD, Standard Deviation; E, Extraversion; A, Agreeableness; N, Emotionality; C, Conscientiousness; O, Openness; OR, Overall Risk; S, Severity; P, Possibility; U, Uncontrollability; ANX, Anxiety; I, Injury; D, Disease; PSY, Psychology; NTT, Nutrition; EVM, Environment; * *p* < 0.05; ** *p* < 0.01.

**Table 4 ijerph-19-00445-t004:** Intercorrelations among variables for the total sample (*n* = 664).

	*M*	*SD*	1	2	3	4	5	6	7	8	9	10	11	12	13	14	15
(1) E	3.72	0.72	1														
(2) A	3.91	0.63	0.69 **	1													
(3) N	3.21	0.77	−0.08 *	0.00	1												
(4) C	3.47	0.71	0.55 **	0.63 **	0.06	1											
(5) O	3.58	0.69	0.62 **	0.63 **	0.11 **	0.58 **	1										
(6) Overall Risk	2.46	0.90	0.06	0.01	0.18 **	−0.06	0.08 *	1									
(7) Severity	2.44	0.95	−0.12 **	−0.06	0.08 *	−0.08 *	0.01	0.65 **	1								
(8) Possibility	2.42	0.85	−0.08 *	0.01	0.26 **	−0.07	0.08	0.56 **	0.49 **	1							
(9) Uncontrollability	2.27	0.96	−0.09 *	−0.12 **	0.28 **	−0.03	0.02	0.43 **	0.47 **	0.59 **	1						
(10) Anxiety	2.31	1.04	−0.03	0.05	0.28 **	0.07	0.09 *	0.32 **	0.44 **	0.42 **	0.47 **	1					
(11) Injury	1.80	0.52	−0.03	−0.03	0.15 **	−0.07	−0.01	0.11 **	0.12 **	0.21 **	0.23 **	0.15 **	1				
(12) Disease	1.98	0.53	−0.07	−0.01	0.09 *	−0.03	−0.09 *	−0.05	0.07	0.04	0.06	0.06	0.47 **	1			
(13) Psychology	2.10	0.53	−0.24 **	−0.06	0.30 **	−0.08*	−0.13 **	0.06	0.08 *	0.17 **	0.25 **	0.16 **	0.36 **	0.47 **	1		
(14) Nutrition	1.22	0.42	0.01	−0.03	0.11 **	0.05	−0.01	0.20 **	0.04	0.06	0.09*	0.01	0.32 **	0.25 **	0.22 **	1	
(15) Environment	1.84	0.44	0.06	−0.04	0.13 **	−0.01	−0.03	0.15 **	0.04	0.10 *	0.11 **	0.01	0.28 **	0.36 **	0.21 **	0.35 **	1

NOTE:M, Mean; SD, Standard Deviation; E, Extraversion; A, Agreeableness; N, Emotionality; C, Conscientiousness; O, Openness; * *p* < 0.05; ** *p* < 0.01.

**Table 5 ijerph-19-00445-t005:** Regression model and variance analysis of different risk events.

Model	*R*	*R* ^2^	Δ*R*^2^	*SE*	*D-W*	*F*	*p*
Injury	0.23	0.10	0.09	0.50	1.92	8.17	0.00 **
Disease	0.14	0.02	0.02	0.53	1.56	6.58	0.01 *
Psychology	0.47	0.22	0.22	0.47	1.81	9.74	0.00 **
Nutrition	0.32	0.10	0.10	0.40	2.00	7.72	0.01 **
Environment	0.26	0.07	0.06	0.42	1.71	8.85	0.00 **

* *p* < 0.05; ** *p* < 0.01.

**Table 6 ijerph-19-00445-t006:** Multiple regression results of prediction of various risk events (*n* = 664).

Dependent Variable	Predictive Variable	*B*	*SEB*	*Beta*	*t*	*p*	*tolerance*	*VIF*
Injury		1.17	0.09		12.61	0.00 **		
	Uncontrollability	0.07	0.03	0.12	2.60	0.01 **	0.61	1.64
	Emotionality	0.07	0.03	0.10	2.51	0.01 *	0.90	1.11
	Body Wight Graded Sports	0.18	0.05	0.14	3.53	0.00 **	0.90	1.11
	Volleyball	0.17	0.05	0.13	3.23	0.00 **	0.92	1.09
	Possibility	0.08	0.03	0.13	2.86	0.00 **	0.63	1.60
Disease		2.02	−0.13		15.10	0.00 **		
	Emotionality	0.07	0.03	0.11	2.71	0.01 **	0.99	1.01
	Openness	−0.08	0.03	−0.10	−2.57	0.01 *	0.99	1.01
Psychology		1.61	0.15		10.77	0.00 **		
	Emotionality	0.17	0.03	0.24	6.69	0.00 **	0.90	1.11
	Extraversion	−0.25	0.04	−0.33	−6.95	0.00 **	0.52	1.93
	Uncontrollability	0.10	0.02	0.18	4.95	0.00 **	0.88	1.11
	Volleyball	0.24	0.05	0.17	4.74	0.00 **	0.89	1.12
	Agreeableness	0.15	0.04	0.17	3.52	0.00 **	0.50	2.00
	Body Wight Graded Sports	0.15	0.05	0.11	3.12	0.00 **	0.09	0.11
Nutrition		1.07	0.05		22.21	0.00 **		
	Overall Risk	0.15	0.02	0.32	6.56	0.00 **	0.56	1.78
	Soccer	−0.24	0.04	−0.22	−5.82	0.00 **	0.92	1.08
	Severity	−0.07	0.02	−0.15	−3.01	0.00 **	0.57	1.75
	Body Wight Graded Sports	−0.11	0.04	−0.11	−2.78	0.01 **	0.90	1.11
Environment		1.59	0.08		19.73	0.00 **		
	Track and Field	−0.19	0.04	−0.19	−4.74	0.00 **	0.92	1.09
	Emotionality	0.06	0.02	0.10	2.48	0.01 *	0.93	1.08
	Overall Risk	0.06	0.02	0.12	3.04	0.00 **	0.95	1.05
	Soccer	−0.13	0.04	−0.12	−2.97	0.00 **	0.89	1.12

* *p* < 0.05; ** *p* < 0.01.

## Data Availability

All relevant data is contained within the article.

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
