# Peer review of "Relationships between Risk Events, Personality Traits, and Risk Perception of Adolescent Athletes in Sports Training"

_ijerph, 2021, doi:10.3390/ijerph19010445_

Round 1
Reviewer 1 Report
have read your research with interest. Thank you.
I have made some comments below.
1. Introduction
- Please describe a previous study of risk events, personality traits, and risk perception of adolescent athletes.
2. Materials and Methods Section
Participants and procedures
- Please describe the criteria for exclusion of study subjects
3. Discussion
-Please describe the limitations of the study.
-Please describe what you would like to suggest in the future as a result of this study.
Reviewer 2 Report
The aim of the article is to investigate the relationship between certain individual characteristics, measured by questionnaires, and the prediction of risk events. Although in general the study deals with interesting topics, the structuring and organisation of the study makes it somewhat confusing and frankly, at times, unreadable. For this reason I recommend a substantial revision before it can be considered for publication.
1) Attention to punctuation, e.g. in the abstract (line 18);
2) Citations in the text, specifically numbers, are sometimes superscripted, sometimes not. Use the same format throughout the text;
3) The socio-descriptive description of the sample is lacking. Use only frequencies for readability, not absolute frequencies. In addition, the part concerning informed consent is missing, i.e. the ethical aspect. Also, what was the average age of the young people interviewed?
4) Rows 102-105: more than a list of demographic information done in this way, it would be more useful to because thisity, this sub-section inserted in this way does not make much sense;
5) Choose in the description of the results (Cronbach's) either the two-digit or the three-digit approximation, not both, and keep it constant in the ENTIRE document;
6) Regarding the ATREQ-A questionnaire, it is necessary to insert the reliability for the subscales, since the analyses are based on them;
7) Table 1 is frankly illegible, it gives you a headache just looking at it. Please amend it so that it is understandable. Also put mean and SD in two separate columns (same for Table 2).
8) Table 3: avoid carriage return in rows where variables are defined;
9) Completely missing any mention of limits.
10) Insert asymmetry and kurtosis of measures.
Round 2
Reviewer 2 Report
Thank you for considering my suggestions.
